# Identification and Validation of Aerobic Adaptation QTLs in Upland Rice

**DOI:** 10.3390/life10050065

**Published:** 2020-05-14

**Authors:** Peng Xu, Jun Yang, Zhenbing Ma, Diqiu Yu, Jiawu Zhou, Dayun Tao, Zichao Li

**Affiliations:** 1State Key Laboratory of Agrobiotechnology/Beijing Key Laboratory of Crop Genetic Improvement, College of Agronomy and Biotechnology, China Agricultural University, Beijing 100193, China; xupeng@xtbg.ac.cn; 2Key Laboratory of Tropical Plant Resources and Sustainable Use, Xishuangbanna Tropical Botanical Garden, Chinese Academy of Sciences, Kunming 650223, China; yangjun@xtbg.org.cn (J.Y.); mazhenbing@xtbg.ac.cn (Z.M.); ydq@xtbg.ac.cn (D.Y.); 3Yunnan Key Laboratory for Rice Genetic Improvement, Food Crops Research Institute, Yunnan Academy of Agricultural Sciences, Kunming 650200, China; zhjiawu@aliyun.com

**Keywords:** upland rice, aerobic adaptation, introgression lines, QTL near-isogenic lines, breeding

## Abstract

The aerobic adaptation of upland rice is considered as the key genetic difference between upland rice and lowland rice. Genetic dissection of the aerobic adaptation is important as the basis for improving drought tolerance and terrestrial adaptation by using the upland rice. We raised BC_1_-BC_3_ introgression lines (ILs) in lowland rice Minghui 63 (MH63) background. The QTLs of yield and yield-related traits were detected based on ILs under the aerobic and lowland environments, and then the yield-related QTLs were identified in a backcrossed inbred population of BC_4_F_5_ under aerobic condition. We further verified phenotypes of QTL near-isogenic lines. Finally, three QTLs responsible for increasing yield in aerobic environment were detected by multiple locations and generations, which were designated as *qAER1*, *qAER3*, and *qAER9* (QTL of aerobic adaptation). The *qAER1* and *qAER9* were fine-mapped. We found that *qAER1* and *qAER9* controlled plant height and heading date, respectively; while both of them increased yields simultaneously by suitable plant height and heading date without delay in the aerobic environment. The phenotypic differences between lowland rice and upland rice in the aerobic environment further supported the above results. We pyramided the two QTLs as corresponding molecular modules in the irrigated lowland rice MH63 background, and successfully developed a new upland rice variety named as Zhongkexilu 2. This study will lay the foundation for using aerobic adaptation QTLs in rice breeding programs and for further cloning the key genes involved in aerobic adaptation.

## 1. Introduction

Rice (*Oryza sativa* L.) as a staple food has fed almost half of the world population, but there are many challenges in rice production because of notably global climate change and water deficit. So far, drought stress has become the main limited factor for rice production [1]. To cope with the shortage of water resources, breeders aim to increase or maintain rice yields during drought periods or in arid areas by using natural genetic variation [2]. Rice including upland rice and lowland rice ecotypes has separately adapted to either aerobic or paddy cultivation. The upland rice represents a predominant ecotype grown under aerobic and rain-fed conditions in the mountainous areas of Southwest China, South and Southeast Asia, Africa, and Latin America, and have been widely adopted in these areas as its low water requirement [3]. Upland rice grows in hydrological conditions like those of other upland crops such as wheat and maize, where the soils are “aerobic” [4]. Accordingly, extensive genetic variations exist in upland rice germplasms for aerobic adaptation [5].

It is undoubtedly that theGreen Revolution breeding of lowland rice has made remarkable achievements. In particular, the development of semi-dwarf, high-yielding, lodging-resistant, fertilizer-responsive rice varieties have provided yield gains and food security for rice consumers over recent decades [6]. However, about 45% of rice-growing areas in rain-fed with direct-seeding environments and lowland rice face the threat of increasing water deficit [7]. Worryingly, little attention has been paid to the improvement of direct-seeding rain-fed upland rice [8]. What is more, the improved rice varieties developed for the paddy field environment are not adaptable for the aerobic upland environment [9].

With labor shortage, changing climate, and water depletion, the potential contribution of direct-seeding rain-fed upland rice, breeding rice varieties with aerobic adaptation has become increasingly more urgent than ever before [10]. However, the current challenge lies in deciphering the genetic basis of the aerobic adaptation, combining that adaptation of upland rice with modern improved rice varieties, thereby breeding high-yield, high-quality, and aerobic-adapted rice varieties [11].

Whether the upland rice was domesticated from wild rice directly to upland rice or from wild rice to lowland rice and then to the upland remains to be controversial. During its long-term selection and domestication, rice has been eventually transformed from requiring marshy or paddy field conditions into the forms adapted to rain-fed upland environments [12]. Adaptation from the aquatic to the aerobic environment may be the main characteristic during domestication of upland rice. The evolution of gene-regulatory sequences was considered as the primary driver of morphological variation [13]. Therefore, distinguishing the different phenotypes and dissecting the corresponding genomic variations underlying the aerobic adaptation of upland rice forms the basis of breeding and development of improved cultivars. 

Recently, a great number of studies have explored the phenotypes and genomic variations between the lowland rice and upland rice, including high genetic variability in the characteristics of morphology and physiology [14], root related traits [15,16,17,18], early growth potential [19,20], osmotic pressure [21,22], and stomata morphology [23]. Among these traits, most of the upland rice varieties exhibited a robust root system, which might be the key physiological characteristics of breeding [24]. To date, besides a large number of identified root-related quantitative trait loci (QTLs), the roles of more and more genes, such as *OsbHLH120* controlling root thickness [25], *OsNAC10* improving drought tolerance and grain yield [26], *DRO1* influencing deep roots by controlling root angle [27], and *SOR1* mutation causing root gravity response and insufficient rooting on soil surface [28], have been elucidated. Although the relevance of root traits in water uptake and drought adaptation was generally accepted [5,17,27,29], the relationship between most of these traits and yields under aerobic conditions remains unclear. Crop yields are the major concerns for the breeders. In the past decades, great progress has been made in detecting large-effect QTLs conferring drought tolerance in both upland rice and lowland rice [30,31,32,33]. However, the detailed phenotype and corresponding genomic variations of upland rice adapted to the aerobic environment still need to be elucidated [5,34]. 

In this study, we manipulated the elite improved upland rice cultivar B6144F-MR-6 as a donor introduced from Indonesia. B6144F-MR-6, widely used in Southeast Asian upland conditions with good aerobic environmental adaptability, has been officially released in Southwest China’s Yunnan Province and Vietnam. Further, B6144F-MR-6 was crossed and backcrossed with the lowland rice cultivar MH63. The BC_1_-BC_3_ upland rice backcross introgression lines (ILs) were obtained, and the yield and intuitive phenotypes of the ILs and their parents were tested under lowland and aerobic rain-fed condition. Aerobic adaptation QTLs were detected and analyzed based on the ILs, and further confirmed in a population of backcross inbred lines (BC_4_ BILs). Moreover, the yield-related traits in the aerobic environment were identified, and two major aerobic adaptation QTLs were identified and located by fine mapping. Finally, a new upland rice variety named as Zhongkexilu 2 was bred in a lowland rice background by molecular breeding methods that introduced clear QTLs as a molecular module.

## 2. Materials and Methods 

### 2.1. Introgression of Aerobic Adaptation from the Upland Rice to Lowland Rice

To understand the effect of introgression from the upland to lowland rice, the elite improved upland rice cultivar B6144F-MR-6 was used as the donor and the high-yield lowland variety MH63 as the recurrent parent in an advanced backcross selection program from the 2005 winter season at the Sanya Breeding Station, Sanya, Hannan Province, China. We obtained not less than 150 BC_1_F_1_ individuals, and 111 of which were backcrossed as BC_2_F_1_ family lines. From each BC_2_F_1_ family line, we randomly selected one plant to backcross to produce BC_3_F_1_ family lines. We harvested BC_1_F_1_ as BC_1_F_2_ populations and harvested BC_2_F_1_ and BC_3_F_1_ as BC_2_F_2_ and BC_3_F_2_, respectively, according to their family lines. Crosses and backcrosses were carried out at the Sanya breeding station with three generations per year. By the 2006 winter season, one BC_1_F_4_ population, 111 BC_2_F_3_ and 96 BC_3_F_2_ family lines were obtained. These materials were screened in the dry season in the aerobic environment of the Sanya Breeding Station (SY-DS, November to April) and in the rainy season of the Menglian Breeding Station, Menglian, Yunnan Province, China (ML-RS, mid-May to September) starting in 2007–2008. In SY-DS, mobile sprinkler irrigation facilities were used to maintain a humid aerobic environment. There was sufficient rainfall in ML-RS so that there was no drought stress during the whole growth period; all materials were grown and managed according to the local protocol. Heading date, vegetative vigor, and phenotypic acceptability (PAcp) were assessed in each screening according to the Standard Evaluation System for Rice (International Rice Research Institute, 2002). Visual selection was made based mainly upon PAcp. For duplicate-based screening, 33 BC_1_F_8_, 23 BC_2_F_7_, and 6 BC_3_F_6_ introgression lines (ILs) were selected from the BC_1_ population and BC_2_, BC_3_ family lines, respectively, based upon better PAcp compared with the recurrent parent in the aerobic condition.

### 2.2. Performance of the ILs under Lowland and Aerobic Conditions at Sanya and Menglian

To understand the genetic basis of the aerobic adaptation, the above mentioned ILs of different generations were used to investigate the differences in biomass, yield, harvest index, plant height, and heading dates between the aerobic and lowland condition (SY-DS and ML-RS, respectively) in 2009. All experiments were conducted under both aerobic and lowland condition with the same experimental design and analysis method, and each experiment was performed with three replications with the respective recurrent parent as control. For the aerobic treatment, we used direct sowing with 3–4 seeds per hole, and thinned to one seedling at the three-leaf stage. For the lowland treatment, sowing and transplanting single seedlings were done. Each plot was 2 × 0.75 m^2^, with three rows per plot, and planting density 25 × 20 cm between rows and plants. The five plants in the middle of the middle rows were analyzed for biomass yield, harvest index, heading date, and plant height. Analysis of variance by ANOVA was performed for each trait. Significant differences between ILs and MH63 were determined by multiple comparisons of LSD values compared with the recurrent parent.

### 2.3. QTLs Mapping of Traits Based upon the ILs 

For QTL mapping, the ILs lines were genotyped using 256 SSR markers that were polymorphic between the two parents, which were uniformly distributed throughout the rice genome. Based on the performance of the ILs under lowland and aerobic conditions, the probabilities of markers linked to each trait were scored by a binomial test. The selection in 2006 to 2008 was made for aerobic performance compared with the lowland-type parent MH63, if the introgression rate of a marker was significantly higher than that of the theoretical prediction from the binomial distribution, the marker may be linked with the corresponding phenotype. In order to avoid the high probability of false QTL from multiple tests, a high significance level of 0.0001 was set [35,36].

### 2.4. Identification of QTLs in the BC_4_F_5_ Backcross Inbred Lines 

Based on the grain yield QTL mapping using the ILs, BC_3_F_8_ introgression line IL-U315 (which carried four grain yield QTLs) was used to raise a BC_4_F_5_ population by backcrossing with MH63 in 2010. More than 500 BC_4_F_4_ backcross inbred lines (BILs) were obtained by the single-seed descent method in the 2013 winter season. The leaves of each BC_4_F_4_ individuals were used for genotypic analysis and the corresponding BC_4_F_5_ individual seeds were used for phenotypic evaluation. We randomly selected 238 BC_4_F_5_ BILs for field phenotypic evaluation in the aerobic condition of ML-RS. Experiments were laid out in an alpha lattice design with three replications; each plot was 2 × 0.75 m^2^ with three rows per plot and planting density 25 × 20 cm. The five plants in the middle of the middle rows were used for biomass, yield, heading date, and plant height measurements. The experiment was conducted under the aerobic condition with direct sowing, normal fertilizer and weed management. A total of 52 introgression SSR markers of IL-U135 were used to genotype the BC_4_F_4_ BILs population. Composite interval mapping of QTLs was performed using IciMapping software version 3.2, combined with the phenotypes of BC_4_F_5_.

### 2.5. Planting and Phenotypic Verification of QTLs Near-Isogenic Lines (NILs) 

Base on the BC_4_F_5_ BILs population, NILs of relevant QTLs were selected with target region and less introgression. Phenotype verification of the NILs and recurrent parent MH63 were conducted under both aerobic and lowland conditions with the same evaluation method of ILs. 

### 2.6. Fine-Mapping of Targeted QTLs

The strategy of substitution mapping was used for fine-mapping the targeted QTLs. The targeted single QTL NILs (BC_4_) backcrossed with the recurrent parent MH63, and BC_5_F_2_ populations were cultivated in the lowland field. Homozygous BC_5_F_2_ individuals with different recombination lengths of the QTL region were selected according to genotype, and the phenotype was verified in the aerobic condition of the Menglian Breeding Station.

### 2.7. Pyramiding Aerobic Adaptation Molecular Modules for Breeding

BC_4_ BILs carried two aerobic adaptation QTLs were chosen for breeding by introduced the QTLs into the lowland rice MH63 background. The multi-site trials were carried out in Hainan and Yunnan Provinces of China from 2016 to 2018 to seek provincial approval of new varieties. In Hainan Province, the multi-site trails were carried out in three sites from 2016 to 2017. Since the test breeding lines was only Zhongkexilu 2 and the control variety B6144F-MR-6, the experimental area of each variety was 0.5 hectare. In Yunnan Province, the trials were carried out in five sites from 2017 to 2018; three breeding lines participated in the trial with the upland rice variety Yunlu 140 as control. The trials were performed for three repetitions with a randomized block, each block was 13.34 m^2^ (0.02 mu). All trials were conducted under upland condition. Sowing at each site according to the beginning of the local rainy season, direct sowing were used with 3–4 seeds per hole, and planting density 25 × 20 cm. Water management depended on natural rainfall without irrigation, and normal fertilizer and weed management were manipulated. The yield and related characters were measured. Significant differences among (or between) breeding lines and control were determined.

## 3. Results

### 3.1. Performance of the Introgression Lines under Lowland and Rain-Fed Aerobic Condition 

We produced the introgression lines (ILs) by crossing and backcrossing the elite upland rice cultivar B6144F-MR-6 with lowland rice cultivar MH63. We tested the performance of three different generations of the ILs in lowland and aerobic environments. For the three experiments, analysis of variance (ANOVA) was performed for each trait (Table 1). In the lowland conditions, the same trend was found at both sites. There were no significant differences among the ILs in yield and biomass, but there were significant differences in harvest index and plant height, except for BC_3_ harvest index at the Menglian site. Heading date showed significant differences at the Sanya site, but not at the Menglian site. In the aerobic conditions, there were significant differences in yield, harvest index and plant height at both sites and in heading date at Sanya, but no significant differences were found in biomass, except for BC_2_ at Menglian. These data indicated that it was possible to introgress aerobic adaptation gene(s) from upland genotype to lowland genotype via backcross and selection, which improved the aerobic adaptation of lowland genotype. The traits with significant differences were analyzed by multiple comparisons with the recurrent parent MH63. Significant and non-significant phenotypes in each generation ILs were divided into phenotypic classes of bimorph and were used for further QTL analysis (Table 1).

### 3.2. QTL Mapping of Traits Based upon the ILs 

Among the ILs that showed significant differences from their recurrent parent MH63, the probabilities of markers linked to particular traits were scored by a binomial test. QTLs were detected for each trait under both aerobic and lowland condition at Sanya and Menglian, respectively (Table 2). In the aerobic condition, a plant height QTL (*qPH1*) was detected on chromosome 1; four heading date QTLs (*qHD3*, *qHD4*, *qHD7*, *qHD12*) were detected on chromosome 3, 4, 7, and 12, respectively. Four grains yield QTLs (*qGY1*, *qGY3*, *qGY7*, *qGY12*) and harvest index QTLs (*qHI1*, *qHI3*, *qHI7*, *qHI12*) were detected in the same region, respectively. In the lowland condition, no QTLs for biomass, yield or harvest index were detected since there were no significant differences between the ILs and the parent MH 63; only a few significant ILs were selected. For plant height, the QTLs detected under irrigation were the same as in the aerobic condition, which indicated that plant height showed consistent genetic influence in both lowland and aerobic condition. Similarly, heading date QTLs were almost the same under irrigation compared with the aerobic condition, except *qHD4*. Thus, QTLs for different traits varied among sites, condition, and generations, but the QTLs for grain yield in the aerobic condition were consistent across sites and generations, and overlapped with plant height and heading date QTLs in the aerobic condition. These results suggest that plant height and heading date were the important traits for aerobic adaptation underlying the increase of yield in the aerobic condition. 

### 3.3. Identification of QTLs in the BC_4_F_5_ Backcross Inbred Lines 

The line IL-U315 (BC_3_F_8_) carried four grain yield QTLs was used to produce the BC_4_F_5_ BILs population (n = 238). Phenotypic variations of the BILs population and the parents were measured for grain yield, plant height, and heading date under aerobic conditions. The descriptive statistics of phenotypic variation were shown in Table 3. Significant phenotypic differences were observed between the parents in this cross; the heading date of IL-U135 was earlier than MH63, while the yield and plant height were higher than MH63. The frequency distributions of the traits in the BILs were plotted; and similar continuous bimodal or multimodal distributions were found for heading date, plant height, and yield (Figure 1). The phenotypic correlations showed that grain yield was highly significantly and negatively correlated with heading date (correlation coefficient = −0.578^**^), and highly significantly correlated with plant height (correlation coefficient = 0.45^**^). Combined 52 introgressions and polymorphic SSR markers between IL-U135 and MH 63, we identified QTLs for grain yield, plant height and heading date in the BC_4_F_5_ BILs population. In the RM212–RM543 interval of chromosome 1, QTLs were detected for plant height and grain yield that accounted for 24.14% of plant height phenotypic variations and 18.7% of yield variations. In the RM218–RM232 interval of chromosome 3, QTLs were detected for heading date and grain yield that accounted for 30% and 19.21% of the phenotypic variations, respectively. In the RM444–RM1328 interval of chromosome 9, one QTL that acted on the heading date was detected, accounting for 19.19% of the heading date phenotypic variations; this QTL also acted on the yield, and accounted for 16.71% of the phenotypic variance (Table 3). These results confirmed the two grain yield QTLs on chromosomes 1 and 3 that were detected in the aerobic condition based upon the ILs, and a new QTL was found on chromosome 9. We named them as Aerobic Adaptation QTLs (*qAER1*, *qAER3*, and *qAER9*). 

### 3.4. NILs Planting and Phenotype Evaluation 

Based on the BC_4_F_5_ BILs population, near-isogenic lines (NILs) for Aerobic Adaptation QTLs were selected from the target region with introgression of smaller segments. Homozygous single-QTL NILs for *qAER1*, *qAER9*, double-QTL pyramided lines for *qAER1* + *qAER9*, *qAER3* + *qAER9*, and triple-QTL pyramided lines *qAER1* + *qAER3* + *qAER9* were screened, but no NILs carried *qAER3* alone or *qAER1* + *qAER3* double-QTL pyramided lines were detected. Phenotypic verification of NILs was conducted under aerobic and lowland conditions. For heading dates, there were no significant difference between all NILs and MH63 except for *qAER1* + *qAER9* in lowland condition, but all NILs were significantly earlier than MH63. For plant height, the NILs carried *qAER1* was significantly higher than that of MH63, *qAER9*, and *qAER3* + *qAER9* in both environments. Similarly, Maximum tillers number per plant of the NILs carried *qAER1* was significantly lower than MH63 in both conditions. Interestingly, MH63 maximum tillers number was more in aerobic condition than in lowland. The effective panicles number was consistent with maximum tillers number, but all NILs was significantly more than MH63. The pyramided lines *qAER1* + *qAER3* + *qAER9* showed more spikelets per panicle in both conditions. There was no significant difference between near isogenic lines and MH63 for the biomass in lowland condition. The NILs carried *qAER1* + *qAER3* + *qAER9*, *qAER3* + *qAER9*, and *qAER1* were significantly higher than MH63 in aerobic condition. The grain setting percentage, harvest index, and yield showed consistent performance; there was no significant difference between NILs and MH63 in lowland, but were significantly higher than MH63 in aerobic condition. We compared QTLs NILs in both lowland and aerobic environments and further verified the phenotype of QTLs. The results indicated that these QTLs increased yield by affecting the plant height (appropriate plant type), not delaying the heading dates, and increased the productive tiller percentage, grain setting percentage, and harvest index in aerobic environment (Figure 2).

### 3.5. Fine-Mapping of the Targeted QTLs

We carried out fine mapping for two main effects-QTLs (*qAER1*, *qAER9*). A NIL for *qAER1* carried a 6 cM integration segment (RM6333–RM12045) was backcrossed with parent MH63 to produce the BC_5_F_2_ population. Based on 3210 BC_5_F_2_ individuals, nine polymorphic SSR markers were used to investigate individual genotypes. Eight homozygous lines (BC_5_F_3_) with different length overlapping segments were screened for phenotypic investigation under aerobic condition (Figure 3a). The phenotypes of the overlapping segment lines showed that lines L5, L6, and L7 had significantly higher plant height and grain yield, while the other lines showed the same phenotype as the parent MH63 (Figure 3b,c). By substitution mapping, *qAER1* was fine-mapped into a 134-kb region between RM11974 and RM11982 (Figure 3a). 

Using the same strategy, the *qAER9* NIL was backcrossed with MH63 to produce a BC_5_F_2_ population. Based on 940 BC_5_F_2_ individuals, eight polymorphic SSR markers were used to investigate individual genotypes in the *qAER9* interval (RM444–RM1328, 13.1 cM), and five BC_5_F_3_ homozygous lines with overlapping segments covered the entire *qAER9* interval were screened (Figure 4a). Phenotypic analysis of all the overlapping lines grown in the aerobic condition showed that the heading date of overlapping lines R3, R4, and R5 were earlier than MH63; the effective panicles, grain setting percentage and grain yield were higher than MH63, while the phenotypes of other overlapping lines were the same as that of MH63 (Figure 4b–e). The three overlapping lines (R3, R4, R5) all carried the introgression marker RM105. We narrowed down and fine-mapped the interval to flanking markers RM5526–RM23966 with a length of 3.0 cM. Further investigations of the phenotype of *qAER9* showed that it resulted in no delay of heading date, but increased the effective panicles, setting percentage and grain yield.

### 3.6. Pyramiding the Aerobic Adaptation Molecular Modules for Breeding

To verify the usefulness of *qAER1* and *qAER9* in upland rice breeding, BC_4_ QTL pyramided line U272, which carried two aerobic adaptation QTLs (*qAER1* + *qAER9*), was chosen for breeding. The multi-site trials were carried out from 2016 to 2018 in Hainan and Yunnan Province of China under the provincial approval system for new upland varieties. The results showed that the yield in Hainan and Yunnan was significantly higher than that of the control (Figure 5a,b). We named the new variety as Zhongkexilu 2, which has been officially approved in Hainan (Approval Number: Qiongshendao 2018033), and the approval in Yunnan is pending. 

## 4. Discussion

### 4.1. Aerobic Adaptation of Upland Rice

During the origin and domestication of rice, human selection, and adaptation to diverse environments have resulted in different ecotypes. Rice is grown under diverse growing conditions such as lowland, rain-fed lowland, rain-fed upland, and flood-prone ecosystems [37]. Thus, the lowland rice and upland rice domesticated as the two major ecotypes of the Asian cultivated rice. In the process of domestication from aquatic (paludal) wild rice or paddy rice to terrestrial upland rice, the growth and development process should be consistent with the soil water supply to avoid drought, and ensure that the water-sensitive reproductive stage is consistent with the rainy season peak through photoperiod control. History and present application areas of upland rice in the Southwest China, Southeast Asia, and South Asia are mainly distributed in the moist upland area. The growth period coincides with the rainy season of the monsoon climate. Drought tolerance may not be the first driving force of upland rice domestication, but may be a second force. Therefore, we suggest that the aerobic adaptation of upland rice is the key genetic difference between the upland rice and lowland rice. Other studies have put forward similar views [11]. 

### 4.2. Phenotype Differences between the Lowland Rice and Upland Rice in the Aerobic Environment 

How to define the phenotype variations adaptation to the aerobic environment between the upland rice and lowland rice is the key to dissecting the aerobic adaptation of upland rice. Those two ecotypes of rice have high genetic variations in morphological and physiological traits [11,14]. However, most studies focus on the difference of drought resistance (tolerance) between lowland and upland rice, and try to design the selection criteria of drought resistance by understanding these drought-related traits [38,39,40], or use marker-assisted selection (MAS) to improve breeding efficiency [41]. In fact, whether using drought-related traits as indicators or using MAS to improve rice drought resistance in breeding practice is not clearly clarified. As the relationship between drought-related traits and yield is not clear, their effects on rice yield under the aerobic environment still limit [42]. Moreover, the expression of major yield QTLs found under drought stress was not consistent in different genetic backgrounds and environments [43]. The aerobic rice breeding progress is still jogged [34]. Comparison the phenotypic differences of a series of typical upland rice and lowland rice verities in aerobic and no-aerobic environments to avoid the complex drought stress, we found that dwarfing plant height and delayed growth period of paddy rice varieties were intuitive phenotypes in the aerobic environment, while the phenotypic differences of typical upland rice varieties in both environments were invisible. Similar results have also been highlighted by some studies [6,8]. In our research, we confirmed two aerobic adaptation QTLs by multiple locations and generations in the aerobic environments without water stress, respectively, acting on plant height and heading date and increasing yield simultaneously in the aerobic environment. These results were consistent with the observed phenotypes.

### 4.3. Does the Green Revolution Gene SD1 Play an Important Role in the Aerobic Adaptation of Upland Rice?

Interestingly, the *qAER1* region contains the so-called Green Revolution dwarfing gene *SD1*. Previous study has comprehensively analyzed co-located QTLs for yield and other traits under drought stress [44]. The region of *SD1* was considered to be the key to controlling drought resistance or aerobic adaptation of upland rice [45,46,47,48]. The Green Revolution involved the widespread adoption of semi-dwarf rice varieties carried *sd1*, and produced more than doubled the global rice yields [49,50]. The high yield potential of modern semi-dwarf varieties was mainly attributed to their improved harvest index, lodging resistance and response to high inputs (mainly nitrogen and water) [51,52]. But most modern semi-dwarf (*sd1*) rice cultivars performed poorly in the rain-fed ecosystems of Asia and Africa [6,53]. Recent studies have shown that *SD1* haplotypes, unique to deep-water rice, promoted internode elongation and the plants adaptation to flooding conditions when submerged. Evolutionary analysis showed that the deep-water rice-specific haplotype was derived from standing variation in wild rice and selected for deep-water rice cultivation [54]. Study of *japonica* rice domestication also proved that the *SD1* locus has been a target of human-mediated selection since prehistoric times [55]. The Green Revolution gene *sd1* related to the plant height of rice; meanwhile, the yield and other traits were also affected [6]. Previous studies have shown that there were abundant genomic variations in *SD1* [56,57]. During the long-term domestication of different eco-types of rice, selection of different haplotypes of *SD1* led to different ecological adaptability. For example, “Green Revolution” haplotypes adapted to a high-input paddy field environment, and deep-water rice haplotypes adapted to the rapid elongation of internodes in a flooded environment. Building on the fine mapping of *qAER1*, we will further analyze the functions of *SD1* haplotypes and other candidate genes in the *qAER1* region of upland rice in the aerobic condition.

### 4.4. The Yield of Plants Largely Depends on the Successful Transition from Vegetative Growth to Reproductive Growth

Flowering is not only an indispensable part of reproduction but also a key stage of development that is vulnerable to environmental stress. However, it is becoming increasingly clear that changing flowering time is an evolutionary strategy for maximizing plant reproduction under different pressures [58]. The aerobic environment of upland rice has more complex environmental stress than that of paddy rice, notably a more erratic water supply. The typical upland rice varieties adapt to the aerobic environment by following mechanisms: Dehydration avoidance characterized by significantly higher growth rate and biomass, efficient partitioning characterized by improving harvest index, and drought escape by accelerating heading [40]. In contrast, most paddy rice varieties show poor adaptation to the aerobic environment, with only some spikelets flowering and fruiting, which results in a large number of ineffective tillers and reduces seed setting rate, in turn leading to reduced yield [8]. We identified *qAER9* on chromosome 9, which acted on orderly heading in the aerobic environment, and improved effective panicle and seed setting rate, ultimately increased the yield. Recent studies have shown that the *OsWOX13* gene enhanced rice drought resistance and induced early flowering, and might be a regulator of other rice drought resistance genes [59]. Further study still need to be conducted for elucidation the molecular function and mechanism of *qAER9*. 

Our study detected and validated two aerobic adaptation QTLs in upland rice, and carried out fine mapping. Aerobic adaptation is a complex agronomic trait controlled by multiple genes. The gene regulatory networks for complex traits often present the characteristics of modularization, in which gene expression is regulated by other major genes and their interactions. Combining into a functional unit, such regulatory genes are responsible for the development of related functions and the formation of target traits as a whole [60,61]. Taking the work forward into breeding practice, we pyramided two QTLs as corresponding molecular modules and introduced them into a lowland rice background. In BC_4_, we observed that the pyramided lines had similar yield potential to the lowland rice parents in the upland environment. By submitting to the Hainan and Yunnan provincial upland rice certification procedures, the stable pyramided line became a variety named as Zhongkexilu 2 that has been approved through Hainan Province. Trials in Yunnan Province also showed excellent field performance. To verify the effects of the QTLs in different genetic backgrounds, we have also screened rice varieties mainly used in Myanmar, Bangladesh, and Nepal as the recipients. We have used molecular marker-assisted selection (MAS) to confirm the introduction of the molecular modules into different genetic backgrounds.

## Figures and Tables

**Figure 1 life-10-00065-f001:**
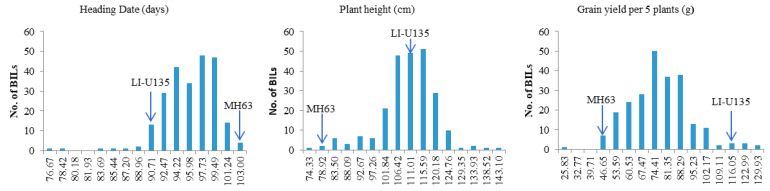
Frequency distributions of the backcross inbred lines (BILs) traits. Parental values were indicated by arrows.

**Figure 2 life-10-00065-f002:**
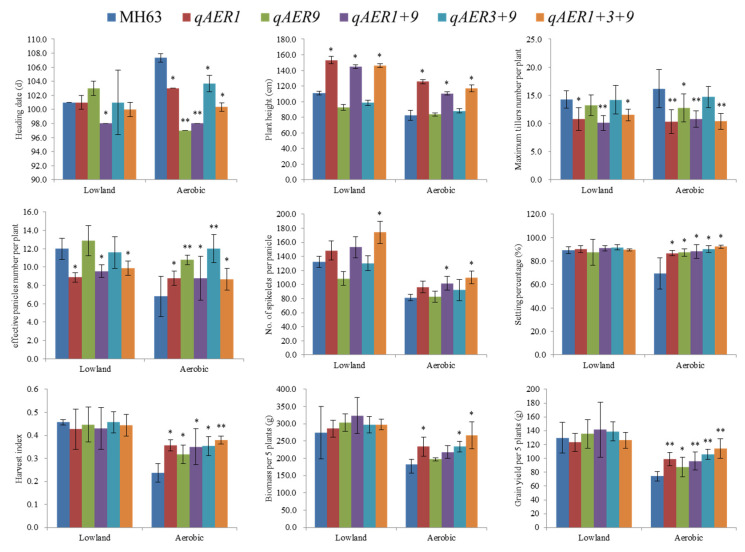
Phenotypic comparison of the NILs with MH63 under aerobic and lowland condition respectively. The asterisk * and ** indicate significant difference from MH63 at *p* < 0.05 and *p* < 0.01, respectively.

**Figure 3 life-10-00065-f003:**
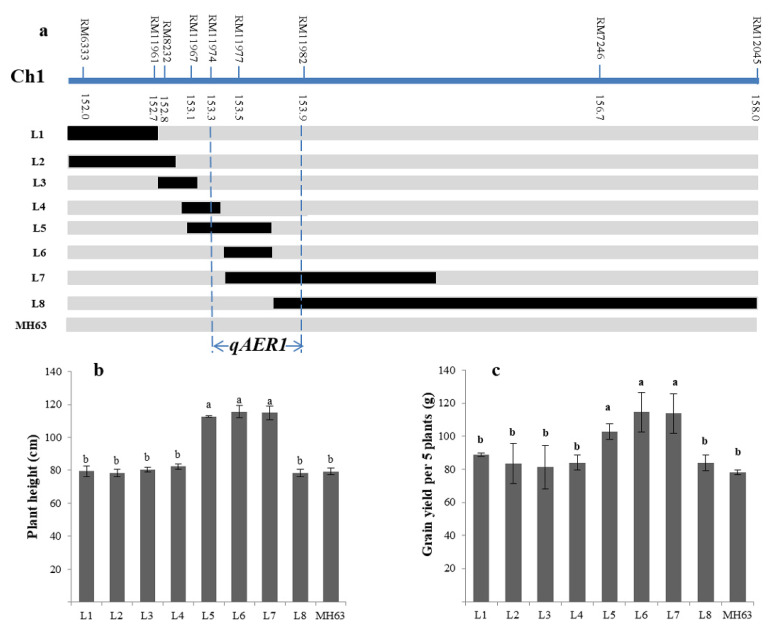
Different length homozygous overlapping segments lines (BC_5_F_3_) of *qAER1* and phenotype evaluation. (**a**) Schematic map of eight recombinants delimiting the mapping region for detailed progeny traits analysis were presented by different bars, and black and grey bars referred to B6144F-MR-6 and MH63 homozygous alleles, respectively. The *qAER1* was mapped to 134-kb region between the markers RM11974 and RM11982 in the long arm of chromosome 1 (Chr1); (**b**) trait comparison of plant height; (**c**) trait comparison of grain yield per 5 plants. Different lowercase letters in the charts indicated significant differences.

**Figure 4 life-10-00065-f004:**
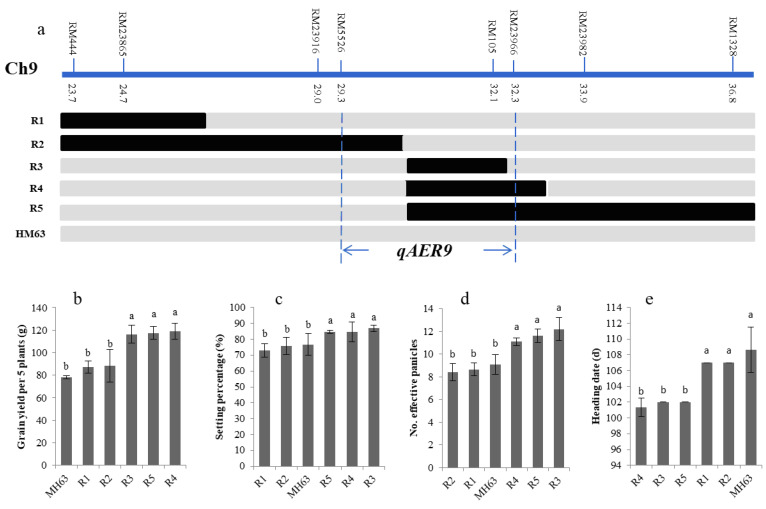
Different length homozygous overlapping segments lines (BC_5_F_3_) of *qAER9* and phenotype evaluation. (**a**) Schematic map of five recombinants delimiting the mapping region for detailed progeny traits analysis is presented by different bars, and the black and grey bars referred to B6144F-MR-6 and MH63 homozygous alleles, respectively; the *qAER9* was mapped to 3.0 cM region between the markers RM5526 and RM23966 in the short arm of chromosome 9 (Chr9); (**b**–**e**) Traits comparison of different length homozygous overlapping segments lines. Different lowercase letters in the charts indicated significant differences.

**Figure 5 life-10-00065-f005:**
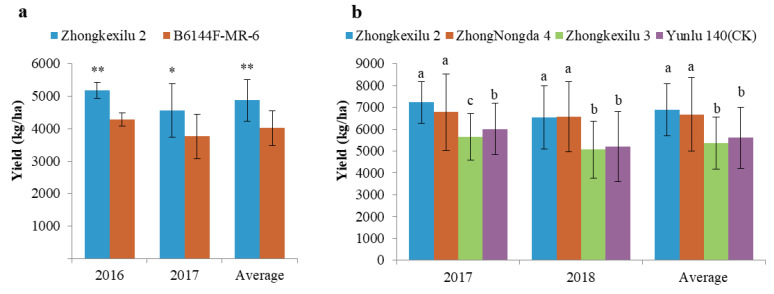
Yield comparison of multi-site trials of the upland rice. (**a**) Hainan Province, Students’ *t*-test was used to assess the differences of yield and determine significant, * *p* < 0.05, ** *p* < 0.01; (**b**) Yunnan Province, significant differences among breeding lines and control were determined by multiple comparisons of LSD values. Different lowercase letters in the charts indicated significant differences.

**Table 1 life-10-00065-t001:** Performance of the introgression lines (ILs) under lowland and aerobic condition at Sanya and Menglian.

Sites and condition	Menglian Breeding Station										Sanya Breeding Station									
	GY		BY		HI		HD		PH		GY		BY		HI		HD		PH	
	L	A	L	A	L	A	L	A	L	A	L	A	L	A	L	A	L	A	L	A
MH 63	52.33	17.37	154.20	134.03	0.34	0.13	99.00	110.50	93.67	61.53	37.00	20.90	114.20	109.90	0.32	0.19	96.00	101.67	95.07	70.47
BC_1_ ILs Mean (N = 33)	60.51	80.97	156.48	230.26	0.39	0.34	97.39	102.58	98.07	97.09	55.19	38.19	138.96	118.25	0.39	0.32	86.77	94.03	119.53	87.15
one-way ANOVA *P*	0.25	0.02	0.65	0.65	0.00	0.00	0.14	0.14	0.00	0.00	0.61	0.05	0.92	0.92	0.00	0.00	0.00	0.00	0.02	0.01
LSD _0.05_	21.68	60.50	47.49	136.55	0.11	0.09	4.25	10.24	5.53	12.55	25.98	19.90	49.21	49.06	0.15	0.09	1.89	3.65	15.58	13.64
Significant lines among 33 BC_1_ ILs		22			2	32			13	33		11				30	31	31	28	27
MH 63	56.48	19.22	162.32	129.87	0.35	0.15	99.00	109.00	93.67	63.25	40.80	22.23	111.20	111.25	0.37	0.20	95.50	102.82	94.80	68.46
BC_2_ ILs Mean (N = 23)	57.90	71.27	161.64	221.74	0.36	0.30	95.14	103.76	106.10	97.02	46.54	34.50	138.79	112.15	0.33	0.31	90.40	93.37	126.66	94.62
one-way ANOVA *P*	0.11	0.00	0.01	0.01	0.00	0.00	0.64	0.64	0.00	0.00	0.46	0.05	1.00	0.99	0.04	0.04	0.00	0.00	0.00	0.00
LSD _0.05_	23.61	57.56	50.53	149.77	0.07	0.07	3.19	14.75	7.93	16.37	27.09	16.85	61.18	41.89	0.12	0.10	2.69	4.06	15.37	9.61
Significant lines among 23 BC_2_ ILs		9		6		23			9	23		6				16	16	23	16	22
MH 63	55.26	18.82	156.26	138.33	0.35	0.14	99.00	109.24	93.67	62.21	36.92	21.22	116.30	109.23	0.32	0.19	96.20	101.33	95.32	69.24
BC_3_ ILs Mean (N = 6)	61.38	51.33	173.45	227.30	0.35	0.22	96.47	109.69	101.05	98.28	43.10	28.56	136.13	119.53	0.31	0.24	88.83	96.17	123.47	92.84
one-way ANOVA *P*	0.05	0.05	0.17	0.17	0.28	0.28	0.09	0.09	0.00	0.00	0.08	0.08	0.91	0.91	0.01	0.01	0.00	0.00	0.00	0.00
LSD _0.05_	30.25	34.79	70.33	132.44	0.06	0.11	4.25	13.49	7.38	20.57	27.01	17.02	47.88	49.19	0.16	0.12	0.93	3.81	17.61	15.90
Significant lines among 6 BC_3_ ILs		5							2	6						1	6	4	5	5

GY: grain yield from 5 plants; BY: biomass yield from 5 plants; HI: harvest index; HD: heading date; PH: plant height; L: lowland condition; A: aerobic condition; ILs: introgression lines.

**Table 2 life-10-00065-t002:** Quantitative trait loci (QTL) mapping of traits based upon the ILs which was significant difference from the recurrent parent MH 63.

QTL	Position (cM)	Flank Markers	*p*	Generation, Site &Condition
*qPH1*	132–147.2	RM297–RM6333	<0.0001	BC_1_ ML A&L, SY A&L; BC_2_ ML A&L, SY A&L; BC_3_ ML L, SY A&L
*qGY1*				BC_1_ ML A, SY A; BC_2_ ML A, SY A; BC_3_ ML A
*qHI1*				BC_1_ ML A, SY A; BC_2_ ML A, SY A
*qHD3*	115.6–127.4	RM2334–RM6329	<0.0001	BC_1_ SY A&L; BC_2_ SY A&L; BC_3_ SY L
*qGY3*				BC_1_ ML A, SY A; BC_2_ ML A, SY A; BC_3_ ML A
*qHI3*				BC_1_ ML A, SY A; BC_2_ ML A, SY A
*qHD4*	3.1–7.9	RM7585–RM5414	<0.0001	BC_1_ SY L; BC_1_ SY L
*qHD7*	93.9–101.8	RM234–RM429	<0.0001	BC_1_ SY A&L; BC_2_ SY A&L
*qGY7*				BC_1_ ML A, SY A; BC_2_ ML A, SY A; BC_3_ ML A
*qHI7*				BC_1_ ML A, SY A; BC_2_ ML A, SY A
*qHD12*	3.2–13.3	RM20–RM6288	<0.0001	BC_1_ SY A&L; BC_2_ SY A&L; BC_3_ SY L
*qGY12*				BC_1_ ML A, SY A; BC_2_ ML A, SY A; BC_3_ ML A
*qHI12*				BC_1_ ML A, SY A; BC_2_ ML A, SY A

*qPH*: QTL of plant height; *qHD*: QTL of heading date; *qGY*: QTL of grain yield; *qHI*: QTL of harvest index; ML: Menglian breeding station; SY: Sanya breeding station; A: aerobic condition; L: lowland condition.

**Table 3 life-10-00065-t003:** Identification of QTLs in the BC_4_F_5_ BILs population.

Traits	Chr	Marker Interval	Position (cM)	LOD	A	*R*^2^ (%)	QTL
Plant height	1	RM212-RM543	135.8–145.6	5.42	12.45	24.14	*qAER1*
Grains yield	1	RM212-RM543	135.8–145.6	4.19	13.87	18.70	
Heading date	3	RM218-RM232	67.8–76.7	6.52	−3.5	30.00	*qAER3*
Grains yield	3	RM218-RM232	67.8–76.7	8.97	16.51	19.19	
Heading date	9	RM218-RM232	23.7–36.8	10.38	−3.04	16.71	*qAER9*
Grains yield	9	RM218-RM232	23.7–36.8	3.18	13.87	16.71	

A: additive effect of allele from upland rice cultivar B6144F-MR-6; R2: proportion of the phenotypic variance explained by the QTL.

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
