# Peer review of "Identification and Validation of Aerobic Adaptation QTLs in Upland Rice"

_life, 2020, doi:10.3390/life10050065_

Round 1
Reviewer 1 Report
In the manuscript titled "Genetic Dissection of Aerobic Adaptation in Upland
Rice: from QTL to Breeding Practice", the authors tried to measure genetic potentials of conceptional "Aerobic Adaptation" character as yield potentials grown in upland condition. They conducted genetic analysis of grain yield, biomass yield, harvest index, heading date, and plant height both at upland and lowland cultivations.
However, it is not sure for the Reviewer whether we can say yield-enhanced QTL just detected on the aerobic condition really say Aerobic Adaptation QTL?
I think this is the most critical point whether the reviewer and reader agree to authors claims that QTLs detected in this study is really Adaptation ?
The authors must show the comparison of yield-associated phenotypes of NILs carrying QTL (qAER1, qAER3, or qAER9) and MH63, grown in aerobic ecosystem and lowland ecosystem.
And difference (or difference of percentage on average) between NIL and MH63 in aerobic environment should be higher than difference (or difference of percentage on average)between between NIL and MH63 in lowland environment. When this data can be show,
we may say this is QTL adaptive on aerobic condition.
Below is an example bar chart to explain my feeling.
---- | NIL, Aerobic
-------------| MH63, Aerobic
---------------------| NIL, lowland
-----------------------| MH63, lowland
Because any yield-enhancing major QTL may become aerobic Adaptation QTL unless evaluation of QTL did not depend on the comparison to result in lowland environment using the same genetic materials.
Please clarify definition of "adaptation" in this manuscript and how we can avoid confusion between Aerobic Adaptation and yield-enhancing QTLs in introduction, materials and methods or results section clearly.
P3, L140
The authors used a binomial test to detect linkage between markers and QTLs.
To apply binomial (or more strictly multinomial) test to detect linkage between markers and phenotypes, the authors would necessary to classify plants showing given phenotypic values to phenotypic classes of bimorph (like tall and short, high-yield and low-yield).
But no one knows each IL always has only one QTL associate to phenotype, so I believe that binomial test is not suitable for detect linkage in ILs.
I recommend to authors that result of ILs should move after explanation of 3.3. Identification of aerobic adaptation QTLs by the BC4F5 backcross inbred lines.
If one IL has more than two QTLs, the theoretical basis of analysis will be collapse.
I checked Citation 60 and 61, but the authors could not follow logic why the authors cited this article to explain the logic in the materials and methods.
Minor comment
The authors used Fisher's LSD for multiple comparisons in the manuscript.
In general the Fisher's LSD did not control type I error (pseudo positive) level due to
multiple tests in multiple comparisons. So the authors should conduct statistical analysis again using Tukey-Krammer or Scheffe multiple comparison instead of LSD.
Therefore this comment may affect to results and discussions depending on the result of statistical test.
According to the comment suggested above, the reviewer suggest to editor " Reconsider after major revision".
Author Response
Dear Reviewer,
We sincerely thank you for your expertise evaluation and specified comments to our manuscript which will definitely help us to improve the quality of the manuscript. We have studied the comments carefully and revised the manuscript following your suggestions. Which we hope can meet with approval. The manuscript was revised using “Track Changes” and highlighted in yellow. The main corrections in the paper and the responses to the reviewer’s comments are as following:
Responds to the reviewer’s comments:
Reviewer #1:
1. In the manuscript titled "Genetic Dissection of Aerobic Adaptation in Upland Rice: from QTL to Breeding Practice", the authors tried to measure genetic potentials of conceptional "Aerobic Adaptation" character as yield potentials grown in upland condition. They conducted genetic analysis of grain yield, biomass yield, harvest index, heading date, and plant height both at upland and lowland cultivations. However, it is not sure for the Reviewer whether we can say yield-enhanced QTL just detected on the aerobic condition really say Aerobic Adaptation QTL? I think this is the most critical point whether the reviewer and reader agree to authors claims that QTLs detected in this study is really Adaptation?
The authors must show the comparison of yield-associated phenotypes of NILs carrying QTL (qAER1, qAER3, or qAER9) and MH63, grown in aerobic ecosystem and lowland ecosystem. And difference (or difference of percentage on average) between NIL and MH63 in aerobic environment should be higher than difference (or difference of percentage on average)between NIL and MH63 in lowland environment. When this data can be show, we may say this is QTL adaptive on aerobic condition.
Below is an example bar chart to explain my feeling.
---- | NIL, Aerobic
-------------| MH63, Aerobic
---------------------| NIL, lowland
-----------------------| MH63, lowland
Because any yield-enhancing major QTL may become aerobic Adaptation QTL unless evaluation of QTL did not depend on the comparison to result in lowland environment using the same genetic materials.
Please clarify definition of "adaptation" in this manuscript and how we can avoid confusion between Aerobic Adaptation and yield-enhancing QTLs in introduction, materials and methods or results section clearly.
Response: Thank you very much for all your comments and suggestions. In our experiment, ILs were used to detect QTLs of yield and yield related traits in the aerobic and lowland environment. Then we used BC4F5 backcrossed recombination inbred lines to identify the yield and yield related traits QTLs in the aerobic environment. Due to the large number of BC4F5 backcrossed recombination inbred lines, we did not evaluated the phenotype at the same time in lowland environment, but verified the phenotype by cultivating QTL near isogenic lines in the aerobic and lowland condition. In the manuscript, we added the results of the comparison of NILs in the aerobic and lowland condition (figure 2 has been modified). The results showed there was no significant difference between NILs and MH63 in lowland, but were significantly increased yield than MH63 in aerobic condition. So we designated as QTL of Aerobic Adaptation. We have made a thorough correction to the revised manuscript including the following mentioned phrases and words. All changes and corrections have been ‘highlighted’.
2. P3, L140
The authors used a binomial test to detect linkage between markers and QTLs. To apply binomial (or more strictly multinomial) test to detect linkage between markers and phenotypes, the authors would necessary to classify plants showing given phenotypic values to phenotypic classes of bimorph (like tall and short, high-yield and low-yield). But no one knows each IL always has only one QTL associate to phenotype, so I believe that binomial test is not suitable for detect linkage in ILs.
Response: Thank you very much for your comments. In our experiment, all ILs were backcrossed with MH63 and selected. We compared the phenotypes of three different generations of the ILs with MH63 in lowland and aerobic environments. Significant and non-significant ILs in each generation was divided into phenotypic classes of bimorph. In theory, it conforms to binomial distribution. Significant and non-significant ILs as two types were used for further QTL analysis.
3. I recommend to authors that result of ILs should move after explanation of 3.4. Identification of aerobic adaptation QTLs by the BC4F5 backcross inbred lines. If one IL has more than two QTLs, the theoretical basis of analysis will be collapse.
Response: We understand the reviewer's concerns. The ILs presented the same phenotype in the same backcross generation, if the introgression rate of one marker is significantly higher than that of the theoretical predication from binomial distribution, the marker may be linked with the corresponding phenotype. In order to avoid the high probability of false QTL from multiple tests, a high significance level of 0.0001 was used. As shown in Table 2. This method can quickly detect the correlation interval in ILs, but it is necessary to verify QTL by the segregation population, so we verified QTLs by the BC4F5 backcross inbred lines population.
4. I checked Citation 60 and 61, but the authors could not follow logic why the authors cited this article to explain the logic in the materials and methods.
Response: We are sorry for the mistakes of order; the references have been reordered and unified the formats. Citation 60 and 61 reorder to [35, 36], these two references contain the discussion and practice of QTL detection by binomial test.
Minor comment
5. The authors used Fisher's LSD for multiple comparisons in the manuscript. In general, the Fisher's LSD did not control type I error (pseudo positive) level due to multiple tests in multiple comparisons. So the authors should conduct statistical analysis again using Tukey-Krammer or Scheffe multiple comparison instead of LSD. Therefore this comment may affect to results and discussions depending on the result of statistical test.
Response: Thank you very much for your advice. In our comparison, only ILs or NILs were compared with rice recurrent parent MH63, and the LSD method was available.
According to the comment suggested above, the reviewer suggest to editor "Reconsider after major revision".
Thank you very much for all your comments and suggestions.
Reviewer 2 Report
The manuscript of the authors aims at deciphering aspects of the aerobic adaptation in upland rice. Their approach is a genetic one.
Aerobic adaptation and drought resistance are necessary for growing rice in “upland”. Upland rice refers to rain fed rice in opposition to rice is usually grown in “lowland” (irrigated or flooded lands) in anaerobic conditions.
Such a study is of interest and clearly aerobic growing is a challenge in rice science. The use QTL to identify genetic locus controlling aerobic adaptation. They use an upland cultivar, B6144F-MR-6, that was crossed and backcrossed with the lowland rice cultivar Minghui 63. Aerobic adaptation and the yield-related traits in the aerobic environment were identified.
The study is interesting. It is scientifically sound, and leads to interesting results. For a biochemist, it is frustrating that the study does not go to the identification if the gene(s) in the QTL of interest. Nevertheless, the study is worth publishing.
I have remarks aiming at clarity of the writing. Authors should think that their paper will be read by people not familiar to QTLs, so they should always ask themselves : is that clear to every reader?
From my point of view, the title sounds more like a bibliographic title. They should use the title to stress out their main message, i.e. the main results, and not the methodology.
While the manuscript is well written, sometimes it is hard to understand because authors do not take cautious to be clear for every reader. The present loci that have not been introduced as though it was obvious they were loci (cf abstract). Abstract should be re-written to be clear for people which have not read the manuscript (which is the aim of an abstract).
Line 19. Maybe an explanation of what is called upland rice and lowland rices should be given.
Line 20: I would replace “by using the” by “in”
Line 20. Do you mean “combining”?
Line 21. What is BC1-BC3
All the sentence is not clear
Line 22. What is BC4F5,
I suggest avoiding such a big and long sentence. Prefer such sentence. “We first…. We then….” Describe the step by step process.
Line 25. What are “qAER1 and qAER9 ».
Line 27. What do you mean by that? How does it corroborate?
Line 28. “pyramided » what does that mean?
Line 62/63: not clear. Maybe delete.
Line 76/ is the grammar correct?
Line 79: elucidated. Genes are not elucidated. Their roles and/or importance can be elucidated.
Lines 90/91. B6144F-MR-6 was first crossed to cultivar Minghui 63 and then the resulting lines were backcrossed with Minghui 63, right?
For non-genetic specialists, can you explain what are BC1 BC2 and BC3?
Lines 93 and 94. What differences do you make between “Aerobic adaptation QTLs” and “
the yield-related traits in the aerobic environment”?
Table I. Introduce another line with ML and SY to be above the traits considered.
Ecosystem: is that the correct term? Cultivation condition sounds better.
Author Response
Dear Reviewer,
We sincerely thank you for your expertise evaluation and specified comments to our manuscript which will definitely help us to improve the quality of the manuscript. We have studied the comments carefully and revised the manuscript following your suggestions. Which we hope can meet with approval. The manuscript was revised using “Track Changes” and highlighted in yellow. The main corrections in the paper and the responses to the reviewer’s comments are as following:
Responds to the reviewer’s comments:
1.The manuscript of the authors aims at deciphering aspects of the aerobic adaptation in upland rice. Their approach is a genetic one. Aerobic adaptation and drought resistance are necessary for growing rice in “upland”. Upland rice refers to rain fed rice in opposition to rice is usually grown in “lowland” (irrigated or flooded lands) in anaerobic conditions. Such a study is of interest and clearly aerobic growing is a challenge in rice science. The use QTL to identify genetic locus controlling aerobic adaptation. They use an upland cultivar, B6144F-MR-6, that was crossed and backcrossed with the lowland rice cultivar Minghui 63. Aerobic adaptation and the yield-related traits in the aerobic environment were identified. The study is interesting. It is scientifically sound, and leads to interesting results. For a biochemist, it is frustrating that the study does not go to the identification if the gene(s) in the QTL of interest. Nevertheless, the study is worth publishing. I have remarks aiming at clarity of the writing. Authors should think that their paper will be read by people not familiar to QTLs, so they should always ask themselves: is that clear to every reader? From my point of view, the title sounds more like a bibliographic title. They should use the title to stress out their main message, i.e. the main results, and not the methodology. While the manuscript is well written, sometimes it is hard to understand because authors do not take cautious to be clear for every reader. The present loci that have not been introduced as though it was obvious they were loci (cf abstract). Abstract should be re-written to be clear for people which have not read the manuscript (which is the aim of an abstract).
Response: Thank you very much for all your comments and suggestions. We have renamed a clearer title, and have made a thorough correction in the revised manuscript including the following mentioned phrases and words. All changes and corrections have been ‘highlighted’.
- Line 19. Maybe an explanation of what is called upland rice and lowland rices should be given.
Response: We sincerely thank the reviewer for this helpful comment. We describe the upland rice and lowland rice in line 48-55.
- Line 20: I would replace “by using the” by “in”
Response: We sincerely thank the reviewer for this helpful comment. The phrase “by using the” is replaced by ‘in’.
- Line 20. Do you mean “combining”?
Response: We sincerely thank you for kind comment. We have modified this sentence.
- Line 21. What is BC1-BC3. All the sentence is not clear.
Response: BC is the abbreviation of terminology of “backcross”, the right subscript number linked to the letter “C” means the backcross times, and the BC1-BC3 mentions the times of backcross generation, e.g, BC3 represents this generation backcrossed with the recurrent parent for 3 times. As shown in other place of the manuscripts, they represent the same meaning.
- Line 22. What is BC4F5,I suggest avoiding such a big and long sentence. Prefer such sentence. “We first…. We then….” Describe the step by step process.
Response: We sincerely thank you for kind comment. The BC4F5 represent this generation backcrossed with the recurrent parent for 4 times, and further inbred for 5 times (“F” means the generation of offspring). As shown in other place of the manuscripts, they represent the same meaning. We have re-written this sentence in “Abstract”.
- Line 25. What are “qAER1 and qAER9».
Response: We sincerely thank you for kind comment. The qAER1 and qAER9 refer to QTLs of Aerobic Adaptation
- Line 27. What do you mean by that? How does it corroborate?
Response: We sincerely thank the reviewer for this helpful comment. “that” is just a conjunctive word used for leading an object clause. We have re-written this sentence as “The observed phenotypic differences between lowland rice and upland rice in the aerobic environment further supported the above results”.
- Line 28. “pyramided » what does that mean?
Response: We sincerely thank you for kind comment. “pyramided” means introducing more than two QTLs into the receptor parent.
- Line 62/63: not clear. Maybe delete.
Response: We sincerely thank you for kind comment. We have modified this sentence for clear.
- Line 76/ is the grammar correct?
Response: We sincerely thank you for kind comment. We have modified this sentence.
- Line 79: elucidated. Genes are not elucidated. Their roles and/or importance can be elucidated.
Response: We sincerely thank you for expertise examination. We have corrected the presentation.
- Lines 90/91. B6144F-MR-6 was first crossed to cultivar Minghui 63 and then the resulting lines were backcrossed with Minghui 63, right? For non-genetic specialists, can you explain what are BC1 BC2 and BC3?
Response: We sincerely thank you for expertise examination. BC is the abbreviation of terminology of “backcross”, the right subscript number linked to the letter “C” means the backcross times, and the BC1-BC3 mentions the times of backcross generation, e.g, BC3 represents this generation backcrossed with the recurrent parent for 3 times.
- Lines 93 and 94. What differences do you make between “Aerobic adaptation QTLs” and “the yield-related traits in the aerobic environment”?
Response: We sincerely thank you for kind comment. In our experiment, ILs were used to detect QTLs of yield and yield related traits in the aerobic and lowland environment. Then we used BC4F5 backcrossed recombination inbred lines to identify the yield and yield related traits QTLs in the aerobic environment. Due to the large number of BC4F5 backcrossed recombination inbred lines, we did not evaluated the phenotype at the same time in lowland environment, but verified the phenotype by cultivating QTL near isogenic lines in the aerobic and lowland condition. Our results showed there were no significant difference between NILs and MH63 in lowland, but were significantly increased yield than MH63 in aerobic condition. Compared with lowland rice, NILs presented obvious aerobic adaptability. So we designated them as Aerobic Adaptation QTLs.
- Table I. Introduce another line with ML and SY to be above the traits considered.
Ecosystem: is that the correct term? Cultivation condition sounds better.
Response: We sincerely thank you for expertise examination. We have revised ‘Table 1’, and the "ecosystem" is replaced by ‘condition’ the manuscript.
Special thanks to you for your good comments.
Round 2
Reviewer 1 Report
After revision, the reviewer agrees to the QTL handled in this study are yield-related QTL adapted in aerobic conditions and the practical approach to know what characters available in the present genetic resources we can use for breeding aerobic conditions.
The modified figure 2 is a very good data representation to appeal this.
The following is the comment in this manuscript.
As a whole claim of the manuscript, It's OK for me. The validated QTLs are confirmed in the subsequent analysis and the existence of QTLs seems not doubtful.
The Fisher's protected LSD is available as a post hoc analysis after ANOVA. But you should not use this in a multiple test (also in multiple comparisons).
Fisher's LSD is a series of pairwise t-tests (equivalent to a multiple test) so adjustment of Familywise error rate is necessary despite multiple comparison or a series of pairwise tests (between MH63 and ILs).
Therefore, "only ILs or NILs were compared with rice recurrent parent MH63, and the LSD method was available" is not justified.
That is why Tukey-Krammer or Dunnet test has been developed.
Please check this post at the same time for your confirmation.
https://stats.stackexchange.com/questions/106367/is-fishers-lsd-as-bad-as-they-say-it-is
Anyway, Please decide this point will be revised or not with your co-authors because QTL validation is conducted well in the subsequent analysis so this is not so serious for your main claim of this manuscript.
In conventional QTL analysis, marker regression analysis is equivalent to a series of ANOVA to t-test. On this point, you will feel this is the multiple test. But in this case, many authors used LOD scores with a permutation test to control family-wise error.